# The Dual Role of Oxidative-Stress-Induced Autophagy in Cellular Senescence: Comprehension and Therapeutic Approaches

**DOI:** 10.3390/antiox12010169

**Published:** 2023-01-11

**Authors:** Pavlos Pantelis, George Theocharous, Nefeli Lagopati, Dimitris Veroutis, Dimitris-Foivos Thanos, Giasemi-Panagiota Lampoglou, Natassa Pippa, Maria-Anna Gatou, Ioanna Tremi, Angelos Papaspyropoulos, Efthymios Kyrodimos, Evangelia A. Pavlatou, Maria Gazouli, Konstantinos Evangelou, Vassilis G. Gorgoulis

**Affiliations:** 1Molecular Carcinogenesis Group, Department of Histology and Embryology, Medical School, National Kapodistrian University of Athens (NKUA), 11527 Athens, Greece; 2Laboratory of Biology, Department of Basic Medical Sciences, Medical School, National and Kapodistrian University of Athens, 11527 Athens, Greece; 3Section of Pharmaceutical Technology, Department of Pharmacy, School of Health Sciences, National and Kapodistrian University of Athens, 15771 Athens, Greece; 4Laboratory of General Chemistry, School of Chemical Engineering, National Technical University of Athens, 6 Zografou Campus, 15789 Zografou, Greece; 5First ENT Department, Hippocration Hospital, University of Athens, 11527 Athens, Greece; 6Biomedical Research Foundation, Academy of Athens, 11527 Athens, Greece; 7Clinical Molecular Pathology, Medical School, University of Dundee, Dundee DD1 9SY, UK; 8Molecular and Clinical Cancer Sciences, Manchester Cancer Research Centre, Manchester Academic Health 19 Sciences Centre, University of Manchester, Manchester M20 4GJ, UK; 9Center for New Biotechnologies and Precision Medicine, Medical School, National and Kapodistrian University of Athens, 11527 Athens, Greece; 10Faculty of Health and Medical Sciences, University of Surrey, Surrey GU2 7YH, UK; 11School of Science and Technology, Hellenic Open University, 26335 Patra, Greece

**Keywords:** oxidative stress, autophagy, cellular senescence, ROS pathways, lipofuscin, SASP, macromolecular damage, macroautophagy

## Abstract

The contemporary lifestyle of the last decade has undeniably caused a tremendous increase in oxidative-stress-inducing environmental sources. This phenomenon is not only connected with the rise of ROS levels in multiple tissues but is also associated with the induction of senescence in different cell types. Several signaling pathways that are associated with the reduction in ROS levels and the regulation of the cell cycle are being activated, so that the organism can battle deleterious effects. Within this context, autophagy plays a significant role. Through autophagy, cells can maintain their homeostasis, as if it were a self-degradation process, which removes the “wounded” molecules from the cells and uses their materials as a substrate for the creation of new useful cell particles. However, the role of autophagy in senescence has both a “dark” and a “bright” side. This review is an attempt to reveal the mechanistic aspects of this dual role. Nanomedicine can play a significant role, providing materials that are able to act by either preventing ROS generation or controllably inducing it, thus functioning as potential therapeutic agents regulating the activation or inhibition of autophagy.

## 1. Introduction

Cells succeed in maintaining a balance in their internal microenvironment through a plethora of mechanisms. When there is a need for a certain process to be carried out, proteins are produced to achieve this goal. These proteins fulfill their purpose by being broken down to the building blocks they originate from, maintaining the balance in that context [1]. However, not everything in a cell runs smoothly at all times. Under specific circumstances, the cell encounters difficulties, which might hinder its well-adjusted nature, its homeostasis. There is a plethora of stimuli that can cause such distress in the cell’s homeostasis, and they are being referred to as stressors. Stressors can constitute several external or internal stimuli, with varying origins (such as UV radiation, genotoxic agents, oncogenes, etc.), which can cause damage to a multitude of the cell’s components, such as proteins, DNA, lipids, organelles, and overall disturb the cell’s homeostasis in various ways, with the most common one being the generation of reactive oxygen species (ROS) [2]. When the cell is required to confront these stressors, it has at its disposal an assortment of different mechanisms that not only attempt to counter the consequences provoked by ROS in cellular molecules referred to above but also aid in the process of purging ROS from the cellular environment [3]. Such mechanisms are the DNA damage repair system [4], proteolytic systems, autophagy, as well as senescence, to a certain extent. 

If the damage inflicted on the cell’s components is irreparable, the cell undergoes a transformation to a state of a generally permanent cell cycle arrest until the said issue is resolved, called senescence. The benefit of senescence is that, since the cell cannot further replicate, the damaged components cannot be passed on through cell division, which is the beneficial side of this process. On the other hand, senescent cells are disfigured, have faulty metabolism and secrete specific factors into their microenvironment, called the senescent-associated secretory phenotype (SASP), which can also promote senescence to their neighboring cells in a paracrine manner, which can ultimately even lead to inflammation and cancer [5,6]. This is where mechanisms such as autophagy are activated in order to contain the lesions and protect the cell from further damage. Autophagy can act chaotically, degrading cell components without specific guidance (macroautophagy), or in a more targeted manner, tracking down and eliminating only damaged components. Either way, this process has the purpose of eradicating the damaged factors in the cell and assisting in returning the cell to its normal state (anti-senescence) [7]. However, in the past years, a different and less studied version of autophagy’s nature has featured in the scientific community’s sights—a version through which autophagy sustains and promotes senescence through specific pathways but also through simply providing the building materials for SASP’s factors. 

The interplay between autophagy and senescence has already been identified. However, the role of autophagy in senescence seems to have both a “dark” and a “bright” side. On the bright side, it has been reported that autophagy battles senescence through elimination of stressors and regulation of proteostasis; meanwhile, on the opposite end, studies have shown a negative aspect of autophagy, promoting the senescence-associated secretory phenotype (SASP).

Nanotechnology can provide several materials that play a key role in the condition of oxidative stress, acting on either preventing ROS from generating or controllably inducing them. Additionally, nanomaterials can promote or inhibit autophagy; herein, they can be considered as potential therapeutic agents, focusing on anticancer treatments or senolytic systems’ development [8]. 

## 2. Cellular Senescence

### 2.1. Fundamentals of Cellular Senescence

Cellular senescence is a biological process implicated in various normal and pathological conditions in embryonic and adult life. In normal embryonic life, senescent cells contribute to tissue development, whereas in adulthood, they contribute to tissue repair [9]. Stress signals can alter the cellular state by causing reversible damage, where the original characteristics of the cell can be restored both on a structural and functional level. In cases of irreversible damage, cells acquire a prolonged, non-proliferative but viable state in order to prevent tissue destruction, known as cellular senescence [10]. Developmental senescence is a conserved mechanism detectable during the reconstruction of embryo regions and structures. In adulthood, the senescence state contributes to tissue homeostasis and repair. Some of the stressors that trigger senescence can be nutrient deprivation, genotoxic factors, irradiation, oncogenes and oxidative stress, mitochondrial and ribosomal dysfunction [11]. 

### 2.2. Hallmarks of Senescence

Senescent cells are defined by a general non-reversible cell cycle arrest, macromolecular damage, deregulated metabolism and secretory phenotype (SASP) [5,12]. This particular cell cycle withdrawal differs from the G_0_ phase and is mediated by p53/p21^WAF1/CIP1^ and p16^INK4A/pRB^ tumor suppressor pathways. Many studies have already described that cell cycle arrest is transient [13]. Under specific conditions, senescent cells escape senescence, acquiring a proliferating and aggressive phenotype able to promote tumorigenesis. Macromolecular damage refers to DNA damage via telomere shortening, genomic instability, protein and lipid peroxidation damage involving ROS, resulting in macromolecular profile alterations, while mitochondria and lysosomes become less functional, leading to deregulated metabolism in general. Senescence-associated secretory phenotype (SASP) has a crucial role as a hallmark of senescence [5]. It consists of a broad spectrum of secreted molecules, including growth factors, cytokines, chemokines and matrix metalloproteinases, and this composition varies depending on the kind of stressor and also the cell type. Moreover, SASP is found to be involved in activation of the immune response, wound healing and tissue plasticity, and it is regulated by different signaling pathways, regulated by transcription factors such as C/EBPβ, NF-κB and GATA4. The interplay of senescent cells via SASP and tissue microenvironment in the context of pathology highlights the clinical aspect of senescent cells as a therapeutic target [12]. SASP mediates immune cell recruitment at the tumor site, but in addition, under a persistent and high intensity, the stressor signals can promote cancer immunosurveillance. The family of interleukins are major components of SASP, promoting tissue inflammation and injury following the MMPs’ activity in tissue remodeling, resulting in disrupted tissue homeostasis. Figure 1 summarizes the hallmarks of senescent cells. 

### 2.3. Lipofuscin Detection

The precise identification of senescent cells is an important issue considering the significant role of senescence in human pathologies [5,11,12,13]. The development of senotherapeutic drugs requires the exact and prompt detection of senescent cells [14]. In terms of the identification of senescence in cells, many factors have been established as senescence markers that can accurately detect and characterize senescent cells (SA-β-Gal, p16^INK4A^, p21^WAF1^, γH2AX). Due to the macromolecular damage, cell cycle arrest and malfunctioning metabolism, senescent cells accumulate lipofuscin within their lysosomes. Lipofuscin is a cellular, non-degradable aggregate, which consists of metals, oxidized proteins and lipids. Lipofuscin accumulation is considered a key attribute of senescent cells [5,15] and can be detected by fluorescence microscopy and flow cytometry, as it is an autofluorescent material [16]. Furthermore, in situ histochemical and cytochemical techniques can be applied for its detection. SenTraGor^TM^, a biotinylated Sudan Black-B (SBB) chemical analog developed by the Gorgoulis lab, reacts with high specificity and sensitivity against lipofuscin. It is widely used for precise detection of senescent cells in any biological material [5,17,18,19,20,21,22]. Various theranostic applications might be developed based on identifying senescent cells through the previously mentioned senescence markers and specifically through lipofuscin detection and also targeting these cells that are characterized as senescent [17,18,19,20,21,22,23] and are related to various pathologies. 

## 3. Oxidative Stress

### 3.1. Oxidative Stress Mechanisms

The disturbance of the balance between the increased generation of ROS and the insufficient action of antioxidant defense systems is known as oxidative stress (OS) [24]. This occurrence is not only aggravated with age but also interferes with multiple tissues’ ability to function normally. This phenomenon has been connected to several chronic illnesses, including diabetes, neurological disorders and cardiovascular diseases that plague older people [25]. OS can occur through organelle disfunctions, where its levels are elevated in relation to the normal amount of ROS that are observed in redox signaling. It is a fact that redox pathways are responsible for controlling the different essential responses of the cell. When ROS strike the mitochondrial membranes and mtDNA, there is a positive feedback mechanism, whereby the mitochondria produce more ROS as a response to this stimulus [26]. This causes mitochondrial dysfunctions, as this molecule can then interact strongly with nuclear proteins, lipids and DNA, resulting in DNA damage and the development of protein adducts. There are different mitochondria-derived ROS (mtROS). Initially, oxygen can be converted to a superoxide anion in two different ways [5,27]. The first depends on xanthine oxidase (XO), while the second refers to mitochondrial respiratory chain complexes I (NADH dehydrogenase) and III (bc1 complex). The superoxide anion is consequently converted into hydrogen peroxide by SOD, and this, in turn, can provide the water, oxygen or hydroxyl radical with glutathione peroxidase, catalase (CAT), thioredoxin peroxidase (TPx) or Fenton reaction, respectively. Oxidative stress can harm cells either through lipid peroxidation of membranes or oxidative modification of proteins. The most severe problem is the outcome of DNA damage that ROS provoke [28]. Oxygen is often released after the oxidation of lipids and is then reduced to water through the respiratory chain of mitochondria. Lipid peroxidation is a multi-step chemical process, which involves various reactions, including initiation, propagation and termination. A hydrogen atom is extracted during initiation by ROS, such as alkoxyl (RO), peroxyl (ROO), hydroxyl (OH) and HO_2_ [29].

Polyunsaturated fatty acids and OH initially combine to create a lipid radical (L), and the latter then interacts with molecular oxygen forming a lipid peroxyl radical (LOO). The LOO species then pick up a hydrogen atom from the nearby fatty acid molecule to create a lipid hydroperoxide (LOOH), which degrades into reactive aldehyde products (LDAs), such as MDA, HNE, ONE, 4-HHE and acrolein, when reduced metals or ascorbate are present. As a consequence, there is a disorder of the main chunk of cell membrane lipids, which causes a variety of alterations. More specifically, there are changes in membrane fluidity and permeability, in the ion transport channels, as well as in the suppression of different metabolic procedures. Termination, which is the last step of the process, includes the creation of a hydroperoxide, which is achieved either through the interaction of a lipid radical (L) with a lipid peroxide (LOO) or through the reaction between a peroxyl radical and tocopherol. This can also occur when two peroxide molecules join to form relatively stable non-reactive species called LOOL or hydroxylated derivatives (LOH), respectively [30].

### 3.2. Oxidative Stress and Macromolecular Damage

It is now understood that the protein residues of histidine, cysteine or lysine may react with 4-HNE to create persistent Michael adducts with a hemiacetal structure [31]. The covalent carbon–carbon bonds with a nucleophile via 1,2- and 1,4-Michael addition reactions are among the biochemical procedures implicated in 4-HNE reactions with proteins [32]. On the other hand, MDA, which is a very abundant aldehyde, reacts with nucleophiles and particularly with Lys residues in order to form Schiff bases. MDA is a key component, which results from the alterations of low-density lipoproteins (LDL). Similarly, Ale’s precursors are fundamental for signal transduction, as they gradually change the structure and the operation of circulating and tissue proteins, with significant consequences for the inflammatory status, cell survival and proliferation [33]. From the above, it becomes clear that these products differ from ROS in the fact that their non-charged form enables them to move across membranes and cytosol with ease and, as a result, to have extensive destructive effects within or outside of cells. This links the MDA and HNE to cancer, age-related neurological disorders and normal aging. 8-oxoGuanine (8-OHG) is an oxidation product produced in the DNA by dG oxidation. Incorrect 8-OHG-to-adenine joining may result in G-T and C-A alterations in the DNA. The nucleoside form of this molecule (8-OHdG) is a marker of oxidative DNA damage, both in vivo and in vitro. 

## 4. Autophagy

### 4.1. Mechanisms of Autophagy

Autophagy, a highly conserved eukaryotic process, is defined as the ability of the cell to maintain its homeostasis under certain conditions through a self-degradation system. Therefore, through the activation of autophagy, it is feasible for cells and organisms to survive during stressful situations, such as infection, starvation and oxidative stress. 

The current scientific data suggest that there are three major systems that are characterized by different systemic procedures: macroautophagy (also mentioned as autophagy), the ubiquitin-proteasome system (UPS) [34] and chaperone-mediated autophagy (CMA). Autophagy or macroautophagy is categorized as the non-selective aspect of autophagy, whereby protein aggregates and malfunctioning organelles are degraded. The main features of macroautophagy are double-membraned vesicle formations, called autophagosomes, acting as mediators in the degradation process through the engulfment of the target molecules. Autophagosome biogenesis is dictated by three processes: initiation, nucleation and elongation (engulfment) of phagophores [35]. Initiation is dictated by the activation of ULK complex and Atg 13. The main intermediary of nucleation is Beclin-1, leading to the engulfment of the targeted cytosolic mass. When the phagophores are formed, they are being transferred to the lysosomes’ location using the cell’s transportation system, where they fuse together and form autophagosomes [36]. Through this process, the toxic mitochondria are eliminated, reducing the oxidative stress load in pathological circumstances. The mTOR pathways have been identified as holding a protagonist role in the activation of macroautophagy, which is dictated by multiple factors [37]. The availability of energy and building blocks appears to be a major factor affecting the mTOR pathway. In an energy-rich state, mTORC1 is over-activated, indirectly lowering the activity of AMPK, leading to autophagy downregulation. On the other hand, in starvation conditions, AMPK is activated, and hence, the activity of mTORC1 is limited, resulting in autophagy upregulation. In addition to the nutrient accessibility factor, there are various stimuli, including stress factors, which can lead to autophagy activation. UPS is precise with its proteolytic function and does not act chaotically. This high specificity is achieved through the use of ubiquitin peptides—protein beacons that act as markers of protein degradation. UPS is compartmentalized by three distinct proteins/protein families that carry out the process of ubiquitination. E1, a Ub activator, E2, a Ub carrier, and E3, a Ub–protein ligase complex, comprise the protein ubiquitination arsenal [34,38,39]. The E3 function results in a multi-Ub tagged substrate, which is further recognized by the 26S proteasome complex. In this machinery, Ub residues are removed and recycled, but the protein substrate is degraded by the protease activity of the proteasome complex maintaining cellular proteostasis [34]. 

UPS is among the major mechanisms of autophagy. It is characterized by high selectivity, since it targets marked cytosolic molecules for degradation [40]. Its primary use is the rapid breakdown of abnormal (unfolded or misfolded) proteins, which are related to a big spectrum of diseases. It has been discovered that UPS has a multi-purpose function; it acts as a promoter of cell cycle by degrading the proteins that impede cell cycle progression, such as inhibitors, thereby allowing the cell to activate otherwise inaccessible processes. In addition to its proliferation-promoting function, UPS also acts as a protection barrier in the processes of oxidized protein elimination and stress response. Furthermore, UPS’s involvement reaches the grounds of signal transduction and gene expression [41]. 

CMA is a form of selective autophagy, which is characterized by lysosomal proteolysis while being vesicle formation independent. CMA’s main role is the maintenance of the cell’s proteostasis by degrading proteins, protein aggregates and malfunctioning organelles. As part of the autophagy arsenal, in case of CMA’s inadequate function, macroautophagy can be deployed. However, macroautophagy underperforms in this area and cannot fully take on and compensate for CMA’s absence [42,43,44]. CMA is dependent on three important factors: HSC70 chaperone, KFERQ-like motifs and LAMP2A lysosomal membrane domain. HSC70, along with other co-chaperones, such as HSC90, recognizes the KFERQ-like motifs on the protein targets on which it binds and allows for transportation to the lysosome’s location, where it interacts with the LAMP2A membrane domain, inducing its trimerization, which results in the formation of a channel through which the target protein enters the lysosome for degradation. 

### 4.2. Autophagy and Cellular Senescence

The interplay between cellular senescence and autophagy has already been identified. The nature of this link, however, still remains a poorly studied field. While both processes are the results of stress accumulation, and they both aim to maintain cell balance, there has not been a clear explanation of the relationship between these two homeostatic systems [45,46]. The first attempts to establish a correlation between the two processes led the investigators to the more obvious conclusion, namely that autophagy is a mechanism that battles senescence via degradation of the damaged components and the elimination of stressors. As the community further investigated the mechanisms behind autophagy, it was becoming apparent that the nature of this procedure is more fluid and not as dogmatic as had been established. Scientists started questioning the one-dimensional nature of this relationship and began accepting a different approach, which describes autophagy as having more of a dual substance. In further detail, the more studied relationship between the two mechanisms is characterized by autophagy acting as a regulator of proteostasis through the removal of stressor stimuli and damaged cell components (damaged DNA, proteins, organelles, etc.), relieving the cell of harm-inducing factors, acting as an anti-senescence mechanism, which has also been proven though inhibition of autophagic mechanisms resulting in accumulation of ROS [45,46,47,48,49]. The current studies show that there are multiple factors that dictate the stressed cell’s fate, which can vary with the cell’s origin as well as internal factor over-expression, such as the caspase inhibitors, which can alter the cell’s response from apoptosis to senescence [46]. On the other end of the spectrum, there have been studies exploring an aspect of autophagy that acts contrary to its already established role, which have shown the existence of a pro-senescence autophagy-mediated function [36]. According to the literature, autophagy is triggered during senescence, promoting SASP through providing the basic building blocks for the production of SASP-related factors [36]. Furthermore, a specific elimination of autophagy-promoting factors, such as autophagy-related 7 and 12, as well as transcription factors in cells has shown to develop characteristics resembling senescence [50]. These contradictory roles of autophagy are schematically represented in Figure 2. 

### 4.3. Pathways Activating Autophagy by ROS

#### 4.3.1. MTOR, AMPK, PI3K/Akt Pathways

ROS can activate different pathways that can lead to the initiation of autophagy (Figure 3). In the first phase of autophagy, ROS may stimulate autophagy by controlling mTOR. The latter is a major negative mechanism of autophagy whose action is affected by multiple signaling pathways, such as the AMPK and PI3K serine/threonine-protein kinase (Akt) pathways [51]. Generally, in the procedure of autophagosome development, ROS can inhibit the operation of Atg4. More specifically, the Atg4 protease splices LC3/Atg8 to create cytoplasmic LC3-I. Then, LC3-I attaches to phosphatidylethanolamine (PE) by a process similar to ubiquitination involving Atg7 and Atg3 (corresponding to E1- and E2-like enzymes, respectively). The lipid version of LC3, sometimes referred to as LC3-II, adheres to the autophagosome membranes. mTOR is a significant regulator of growth factors and a nutrient-sensing kinase, which, along with AMPK, an energy-sensing kinase, regulates autophagy.

AMP-activated protein kinase (AMPK) is responsible for sustaining the homeostasis of cells, as it controls the energy metabolic rate by organizing various metabolic pathways. Studies have revealed that induction of the hypoxia-inducible factor (HIF) is necessary for the triggering of AMPK under hypoxia [52]. Furthermore, autophagy is another mechanism through which AMPK controls the existence of cells under hypoxic stress. According to studies, AMPK stimulation regulates the energy balance by focusing on several significant substrates at the level of several organs. The defense system of autophagy enables cells to endure various stresses [32]. The mammalian homologous protein mTOR may form two complexes, the 1 and 2 mTOR complexes, with differing functional characteristics. By blocking the mTORC1 pathway, AMPK can promote autophagy, which is particularly beneficial in case of a lack of nutrients. AMPK is activated in hypotrophic situations and phosphorylates TSC2, which prevents mTORC1 from functioning. To control the mTOR pathway, AMPK directly phosphorylates the mTOR-associated Raptor protein at Ser722 and Ser792 sites. 

The Raptor protein is coupled to mTOR in one form, known as mTORCl, while the other, mTORC2, is coupled to Rictor. Although mTORC2 is primarily implicated in cell existence and cytoskeleton recombination, mTORCl primarily regulates cell proliferation, apoptosis and autophagy. As a result, mTORCl is crucial in the control of autophagy, and its activation has the opposite impact on that regulation [53]. 

Angiopoietin I (Ang1), insulin, vascular endothelial growth factor (VEGF), human growth factor (HGF), fibroblast growth factor (FGF) and other growth factors and signal transduction complexes can initiate PI3K activation. A corresponding receptor triggers type I Phosphatidylinositol Triphosphate Kinase (PIP3). During this phenomenon, PIP is phosphorylated, forming PIP3, which can bind to the intracellular signaling protein AKT, activating AKT through synergistic action with phosphoinositol-dependent protein kinase I. Class I PI3Ks’ PIP2 and PIP3 interact with Akt, a PI3K downstream effector, to activate mTOR and suppress autophagy. However, mTORC1 activity is influenced by different positive signals (oxygen, energy, growth factor and amino acid levels), which lead to autophagy inhibition. According to the latest investigations, the raptor in mTORC1 is a direct substratum for the phosphorylation of AMPK. In general, the latter, when the intracellular energy state seems irregular, eases the mTORC1-mediated inhibition of autophagy induction [54].

#### 4.3.2. MAPK, ERK, p38, ERK1/2, PTEN and p70S6K Pathways

ROS can also regulate autophagy through the mitogen-activated protein kinase (MAPK) signaling pathway. The p38 kinase, the extracellular signal-regulated kinase (ERK) and the c-Jun amino-terminal kinase (JNK) are components of the MAPK pathway, which are successively activated in a cascade manner. These molecules perform significant roles in various cell procedures. According to recent research, the MAPK pathway can modify the action of the transcription factors, namely the nuclear factor kappa B (NF-kappa B), forkhead box transcription factor O (FoxO) and activator protein 1 (AP-1), which control the genes associated with autophagy and have an impact on this procedure [55]. Moreover, in a positive feedback mechanism, ROS might arise via MAPK-mediated activation of autophagy [55]. The ROS-activated autophagosome/lysosomal fusion stage, which is correlated with the production of Atg7 and E3 during protein ubiquitination, involves the p38 signaling pathway, and this procedure is dependent on the activation of FoxO transcription. 

#### 4.3.3. Oxidative-Stress-Inducing MAPK ERK Pathway in Cells 

Various studies showed that arsenite stimulates autophagy, as it provokes oxidative stress while simultaneously activating oxidative-stress-related signaling pathways (ERK1/2, PTEN and MAPK) in human uroepithelial cells [56,57,58]. More specifically, in this study, it was revealed that arsenite decreased OGG1 while at the same time increasing 8-OHdG and ATF3. This phenomenon induces autophagy in SV-HUC-1 cells. The related experiments showed that arsenite reduced PTEN and activated p70S6K after ERK1/2 phosphorylation as well as DNA hypermethylation. It must be emphasized that these findings correspond with the fact that not only was PTEN shown to be decreased in UC patients from BFD areas but, simultaneously, p70S6K appeared to be increased in them. These findings suggest that oxidative stress, which controls the activation of the PTEN, ERK1/2 and p70S6K signaling pathways, mediates the induction of autophagy, which is caused by arsenite exposure [55,56,57,58].

#### 4.3.4. Induction of Autophagy through JNK Signaling Pathway

Moderate levels of ROS may transiently activate JNK (Jun N-terminal kinase) signaling, which, with the intervention of the Beclin 1 pathway, can induce autophagy. ROS that exceed a certain level lead to a constant stimulation of JNK, inducing apoptosis, which is controlled by mitochondria [59]. There are many examples of the ways in which ROS activate JNK:(A)The Apoptosis Signal-Regulated Kinase 1 (ASK1) is considered a very important enzyme, which is required for the activation of the JNK pathway. After phosphorylation of MKK4 and MKK7, a process influenced by the levels of ROS in the cell, ASK1 activates the MAPK of the JNK pathway.(B)MLK3 phosphorylates and activates the serine/threonine-protein kinase of the MAPK pathway. This protein is the bridge for the connection of ROS with JNK. Additionally, it promotes the upstream activity of this signaling pathway. The small G protein RALA can be triggered by oxidative stress through the c-Jun amino-terminal interacting protein 1 (JIP1) scaffold complex, which, in turn, regulates the phosphorylation of JNK, according to the study of Van den Berg et al. [60].

I Another significant pathway via which ROS activate JNK is the receptor-interacting protein (RIP)–TRAF2 complex pathway. Recent research has demonstrated that TRAF2 and RIP may interact with one another in the cell membranes to generate RIP–TRAF2 signal complexes, and this phenomenon is followed by activation of the JNK pathway when ROS are produced [60].

According to recent studies, when ROS trigger JNK1, there is a direct phosphorylation of Bcl-2 to detach Bcl-2 from Beclin 1. Moreover, Beclin 1 can act as a substrate to produce a Beclin 1-Vps34-PΙ3K multi-protein complex, hence inducing autophagy.

#### 4.3.5. p62/Keap1/Nrf2 System: The Relationship between Autophagy and Redox Response

It is established that the Mit/TFE family of transcription factors is linked to the relationship between ROS and autophagy [17]. Recent research has revealed that members of the Mit/TFE protein family (TFEB Cys212, TFE3 Cys322 and MITF Cys281) include redox-sensitive Cys residues that facilitate a quick reaction to increasing intracellular oxidative stress as they undergo nuclear translocation [61].

Through p62, a redox-sensitive autophagy receptor protein, oxidative stress influences the selectivity of the autophagy process. The development of intermolecular Cys bonds, which are disulphide-linked conjugates, is what helps the assembly of the p62 oligomer form [57]. Importantly, two Cys residues (Cys105 and Cys113) in p62 protein’s regulatory epitope, which are both required for the activation of pro-survival autophagy, were found to be dependent on elevated ROS [33]. 

Recently, it has been revealed that the antioxidant transcription factor Nrf2 (nuclear factor erythroid 2-related factor 2) can be activated by p62 via an “abnormal” mechanism. This underlying process includes the recruitment of Kelch-like ECH-associated protein 1 (Keap1), which serves as an adapter protein of the Cul3-ubiquitin. According to this concept, p62 is phosphorylated at Ser351 and then attaches to clusters of ubiquitylated proteins and improves its affinity for Keap1. As a result of this action, Keap1 is degraded by autophagy, allowing Nrf2 to build up and move around the nucleus freely. Nrf2 binds to the antioxidant-responsive elements (ARE) in the promoter regions of detoxification genes and also genes involved in the response to DNA damage, such as 8-oxoguanine glycosylase (OGG1) and p53 binding protein 1 (53BP1), triggering the transcription of these genes. Sestrins, a highly conserved family of small “antioxidant-like” proteins, which are activated by p53 under the influence of stressors and are involved in autophagy because they act as AMPK activators and maintain Nrf2 activation by this pathway, have also been proposed [62,63,64].

## 5. Therapeutic Approaches

### 5.1. Senescence: Therapeutic Approaches

Senescent cells are considered therapeutic targets, as they play a crucial role in a wide range of pathologies, especially via SASP factors’ release [65]. Since senescent cells are resistant to apoptosis, the discovery of senotherapeutic drugs represents a developing and highly promising field of current research for new therapies. There is a distinction in the classification of senotherapeutic molecules into senolytics and senomorphics.

Senolytics are compounds that selectively eliminate senescent cells and are able to decrease both senescent cells and inflammation while seeming to comfort frailty in humans [65]. Senomorphics are molecules that can inhibit SASP and suppress senescence indirectly [66]. Adjuvant senostatic interventions, which suppress senescence-associated bystander signaling, might also have therapeutic potential [67].

### 5.2. Oxidative Stress: Therapeutic Approaches

As is clear from the above, oxidative stress can accelerate the progress of age-related pathologies and cancer [68]. Consequently, the scientific community must focus on the design of different treatment strategies in order to counter this situation. The current therapeutic regimens include the use of antioxidant molecules, pro-oxidant therapy and application of glycolysis inhibitors. More specifically, after the concentration of an antioxidant compound is settled, so that a successful cellular response can be completed, there is an equilibrium in the levels of ROS in the cell. The pro-oxidant therapy focuses on the employment of in vivo pro-oxidant agents, such as antitumor agents or radical scavengers, which regulate the elevated ROS levels. Moreover, the use of non-hydrolysable glucose analogs could halt the progression of the earliest stages of glycolysis to the ultimate cascade of metabolic breakdown [69].

### 5.3. Autophagy: Therapeutic Approaches

Over the years of studying autophagy, it was discovered that this homeostatic mechanism is very complex and multi-purpose. It extends from simple everyday tasks, such as macromolecule degradation, all the way to defense against several pathologies, including cancer. Unfortunately, uncertainty shrouds the role of autophagy in the context of pro- or anti-cancer function, since its outcome is heavily dependent on several parameters, such as biomarkers’ presence and gene over/under-expression [70]. Deletion of the autophagic system in the very early stages of development has proven lethal to the developing organism, and crippling of the autophagic machinery, such as autophagy-related 5 or -7 (Atg5, Atg7), has proven to give rise to neurodegenerative conditions. In the same manner, the administration of factors that promote autophagy appears to improve the pathological state in diseases such as Alzheimer’s. Pathologies have also been identified, which benefit from autophagy modulation and not necessarily activation. Depending on the stage of the pathology the individual is experiencing, a different approach is recommended regarding treatment through the autophagic system [71].

## 6. The Role of Nanotechnology in Autophagy, Oxidative Stress and Cellular Senescence

### 6.1. Nanomaterials and Induction of Oxidative Stress as a Therapeutic Strategy

The application of nanotechnology in biomedicine is referred to as nanomedicine. Thus, nanomedicine can exploit the fundamentals and the potential of nanotechnology for the prevention, diagnosis and possible treatment of several diseases. Accurate nanosensors [72], targeted drug delivery systems, as well as molecular imaging are among the greatest challenges in nanomedicine [73,74,75]. 

One of the main categories of nanomaterials, that of a semiconductor photocatalyst, focuses on inducing oxidative-stress-mediated cytotoxicity. The mechanism of photocatalysis is well studied. In brief, light carrying an appropriate amount of energy comes across the photocatalyst, overcoming the energy band gap E_g_. Hence, the electrons lying on the valence band (VB) are excited and move to the conductivity band (CB), consequently leaving positive holes in the VB. These charges react with water (H_2_O) and O_2_ molecules around them, generating ROS (Figure 4). 

Although OS is a condition that has generally been connected to several diseases, it may be therapeutically exploited when it is controllably induced. Therefore, photocatalytic nanoparticles, such as titanium dioxide, or composite thermo-responsive titania-based materials [76], zinc oxide [77], etc., can be used as photo-activated anticancer agents to target cancer cells and induce oxidative-stress-mediated apoptosis [73], an improved photodynamic therapy [73]. 

Furthermore, silver nanoparticles have been proven to induce oxidative stress, allowing anticancer effect on MCF-7, HepG2 [78] and HCT-116 [78,79] cell lines. Moreover, gallium nanoparticles have shown anticancer redox potential in mice [80]. Additionally, the anticancer effect of stabilized copper nanoparticles has been reported in several cell lines (A549, A375, C6-G, MCF-7) [81].

Additionally, their potential could be focused on inducing oxidative-stress-mediated cellular senescence [82]. Since OS is a possible “stressor”, leading to senescence in cells, as previously described, then nanoparticles might potentially induce controllable cellular senescence in cancer cells. Consequently, senolytic drugs might possibly be administered via nano-carriers and destroy senescent cells. 

### 6.2. Nanomaterials and Prevention of Oxidative Stress

Antioxidants can inhibit oxidation, blocking the production of ROS. Numerous dietary [83] as well as synthetic antioxidants are reported, acting as effective therapeutic agents [84]. Nano-antioxidants are nanomaterials that are synthesized in order to prevent OS [85,86]. 

Metallic (e.g., gold (Au) [87]) or green synthesized silver nanoparticles [88], bi-metallic [89], metal oxides (e.g., cerium oxide), carbon nanotubes and some categories of organic nanoparticles (e.g., lignin) and polymeric, as well as composite and hybrid nanomaterials, can exhibit antioxidant properties, often by ROS trapping or possessing catalase-like, glutathione-like or superoxide dismutase-like behavior, breaking chains [90], or by binding to antioxidants or just carrying materials with anti-oxidative behavior [91].

Some oxide-based nanoparticles, due to their intrinsic physicochemical properties, can trap the reactive nitrogen and ROS (RNS/ROS), mimicking the antioxidant molecule and acting as scavengers [92]. Cerium oxide (CeO_2_) nanoparticles possess the multi-enzymatic scavenging potential of ROS and present regenerative properties [92]. The utilization of iron oxide for surface functionalization can also enhance the antioxidant potential [73,93]. Nanoparticles based on platinum (Pt) have shown significant capacity in degradation of hydrogen peroxide [94]. Other studies revealed that trimanganese tetraoxide (Mn_3_O_4_) can protect cells from oxidative damage and can thus be considered a potential nano-therapeutic against ROS-related neurodegenerative diseases [95]. Vanadium pentoxide (V_2_O_5_) can restore the redox balance, thus preventing cells from ROS and protecting the organism against cardiac disorders, neurological disorders and ageing [96]. Graphene oxide-selenium (GO-Se) nanoparticles are also supposed to protect cells against ROS [97]. Polyethylene glycol coated magnesium oxide (PEG-MnO) presents antioxidant activity and can be exploited in magnetic resonance imaging [98]. Additionally, platinum [94] and palladium (Pd) nanocrystals can act as antioxidants [99]. Zirconium dioxide (ZrO_2_) is proven to show anticancer potential and high antioxidant activity [100]. PEG-coated melanin can act as a ROS scavenger [101]. Salvianic acid A (SA)-coated gold nanoparticles (SA-Au) [102], silicon dioxide (SiO_2_) nanoparticles functionalized with gallic acid (GA) [103], curcumin linked to halloysite nanotubes (HNTs) [104] also show antioxidant potential, while flavonoids linked to fullerene (C_60_) present chain-breaking antioxidant activity [105]. Generally, nanocomposite materials based on mesoporous silica and organic compounds have also been studied for their antioxidant capacity [106]. 

Thus, various characteristics, such as surface charge, size, crystallinity, coating, etc., can define the antioxidant capacity of those nanomaterials [107]. Some of the aforementioned categories of nano-antioxidants are presented in Figure 5.

### 6.3. Nanomaterials and Autophagy

Autophagy holds a dual role in both cell survival and death, as was previously mentioned. Since cell death is often mediated by nanomaterial-induced toxicity, autophagy is considered an important cell death mechanism [108]. Various nanomaterials, mainly inorganic, can perturb autophagy, and this issue should be considered in biomedical applications [109]. It is evidenced that intracellular nanoparticles can be sequestered by autophagosomes and consequently degraded via the auto-lysosomal pathway. Perhaps this is an intrinsic protective mechanism. However, recent studies indicate that silver nanoparticles (AgNPs) can activate autophagy, but they fail to trigger the lysosomal degradation pathway, which leads to a cytotoxic effect (defective autophagic flux) [110]. Autophagy dysfunction can be exploited for targeted therapeutic approaches. Generally, depending on the scope, activation, inhibition or dysfunction of autophagy using nanostructured agents might be induced [111]. 

A deep knowledge of the molecular mechanisms would be valuable for nanomaterial hazard assessment in cases of applications where biocompatibility is a crucial parameter and also for the estimation of cytotoxicity in cases of therapeutic strategy development. Thus, if the machinery through which nanomaterials perturb biological systems, focusing on the possible induction of autophagy, were totally clarified, then a more comprehensive elucidation of nanotoxicity would be feasible [108].

## 7. Conclusions and Future Perspectives

Various external environmental sources are capable of inducing oxidative stress, which is associated with several pathologies. OS is responsible for the induction of senescence in different cell types. Numerous signaling pathways are mediated by the reduction in ROS levels and regulation of the cell cycle in order to maintain the organism’s homeostasis. Autophagy is a process that helps cells protect their integrity. Autophagy can promote the degradation of any damaged biomolecules by simultaneously forming building blocks for the synthesis of new useful cell components.

It seems that the role of autophagy is dual, presenting a “dark” and a “bright” side, concerning cellular senescence. In particular, it has been reported that autophagy may restrict cellular senescence through the elimination of stressors and regulation of proteostasis. On the contrary, autophagy can promote SASP, affecting either the cell itself or its extracellular environment, thereby perpetuating cellular senescence. 

Innovative approaches for controlling oxidative stress or autophagy, and indirectly, cellular senescence, might include the use of nanomaterials. Various nanoparticles can prevent ROS production, while others can induce oxidative stress in a controlled fashion to selectively target cancer cells. Furthermore, nanomaterials have the potential to act by either inhibiting or activating autophagy, highlighting their potential for being exploited as meaningful therapeutic agents. 

## Figures and Tables

**Figure 1 antioxidants-12-00169-f001:**
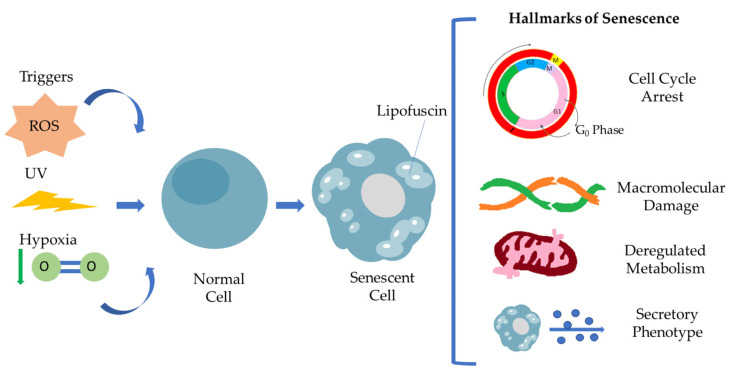
Representation of the hallmarks of cellular senescence. Different stimuli, including ROS, UV irradiation and hypoxia, are able to induce cellular senescence. Senescent cells are characterized by a specific secretory phenotype, cell cycle arrest, macromolecular damage, deregulated metabolism resulting in Lipofuscin accumulation.

**Figure 2 antioxidants-12-00169-f002:**
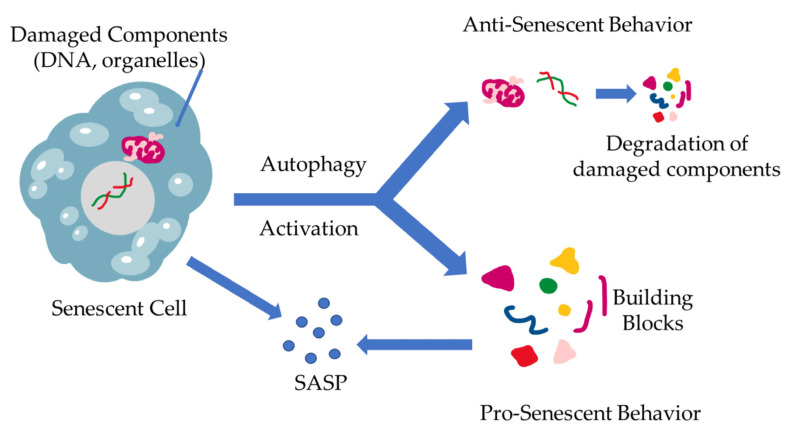
The dual role of autophagy in cellular senescence. When stressors result in damage accumulation in cells, a state of senescence is triggered, leading to SASP production. At the same time, a different fate might befall the cell, leading to the induction of autophagy, eliminating the damaged cells. This results in building block (amino acids, nucleotides, etc.) formation through cell component degradation via autophagy. These building blocks, however, can be utilized by the already senescent cells in SASP production and excretion.

**Figure 3 antioxidants-12-00169-f003:**
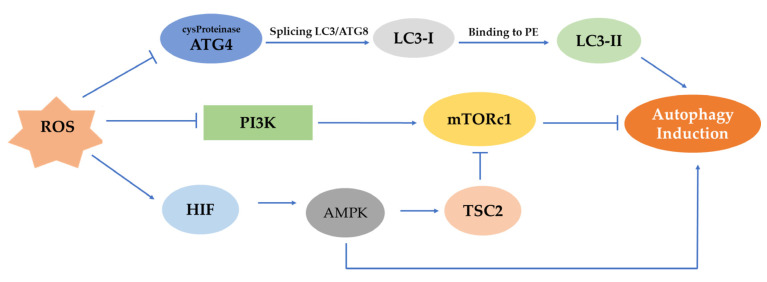
Representation of mTOR, AMPK, PI3K/Akt pathways and their effect on the induction of autophagy. ROS can prevent the activation of ATG4 protease, which splices LC3/Atg8 to produce LC3-I, and this, in turn, after its binding to PE, induces autophagy. In a different mechanism, the AMPK pathway is being activated under hypoxia by HIF, and autophagy can be initiated. However, in the PI3K/Akt pathway, the control of mTORc1 through ROS-mediated inhibition of PI3K can suppress autophagy.

**Figure 4 antioxidants-12-00169-f004:**
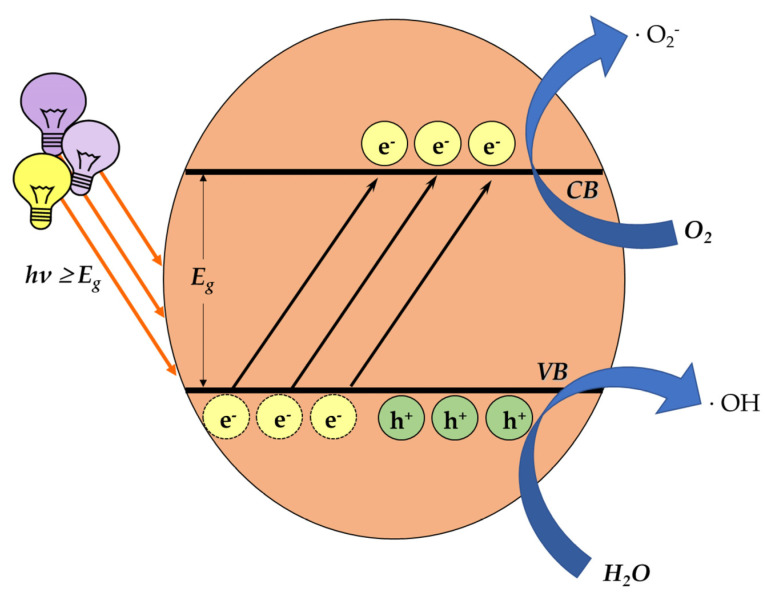
The process of ROS generation through photocatalysis. When light energy is transferred on the surface of a photocatalyst, the excited electrons move from the valence band to the conductivity band, creating positive holes in the valence band. The generated charges, reacting with water (H_2_O) and O_2_ molecules around them, form ROS.

**Figure 5 antioxidants-12-00169-f005:**
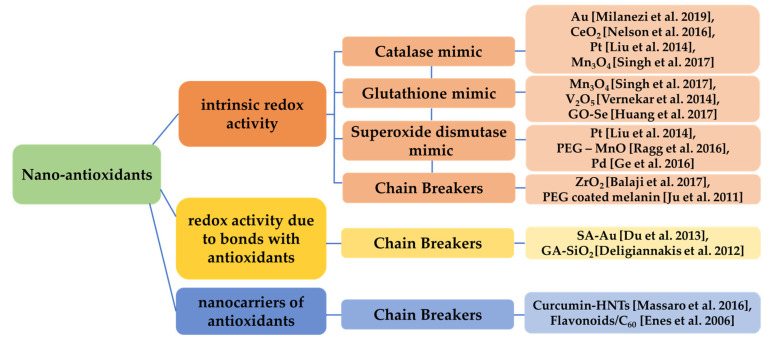
Categorization of nanomaterials acting as antioxidants based on their redox activity. Various nanomaterials present intrinsic redox activity (catalase-like, glutathione-like, superoxide dismutase-like, chain breaking) [87,92,94,95,96,97,98,99,100,101], redox activity related to bonds of the nanomaterial with other antioxidants [102,103] and redox activity due to the fact that act as nanocarriers of antioxidants [104,105].

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
