# Peer review of "The Dual Role of Oxidative-Stress-Induced Autophagy in Cellular Senescence: Comprehension and Therapeutic Approaches"

_antioxidants, 2023, doi:10.3390/antiox12010169_

Round 1
Reviewer 1 Report
The authors provide coverage of the latest developments in oxidative stress-induced autophagy and its interplay with cellular senescence. They also discussed novel nanotechnology approaches for the control of oxidative stress and autophagy and their potential clinical applications. This is a timely and comprehensive review. There are nevertheless several issues that need to be addressed.
Specific comments:
1. Subsection 4.2. It is necessary to elaborate on details of the interaction between autophagy and senescence.
2. Reference 46 is not related to autophagy and senescence. It should be replaced with relevant references.
3. It is important to mention that the DNA damage repair system, autophagy, and senescence not only aid the process of purging ROS from the cellular environment but also deal with the consequences of ROS-mediated damage (The Introduction section, lines 60-63).
4. Figure legends need to be expanded in order to enable the reader to understand figures without having to refer to the main text.
5. The authors have not numbered the different sections and subsections properly (throughout the paper).
Author Response
Response to Reviewer 1 Comments
The authors would like to thank the reviewer for the valuable comments and for giving them the opportunity to revise their manuscript.
All the modifications have been highlighted in yellow in the revised version of the manuscript. The reference and figure numbering has also been changed.
Comments and Suggestions for Authors: The authors provide coverage of the latest developments in oxidative stress-induced autophagy and its interplay with cellular senescence. They also discussed novel nanotechnology approaches for the control of oxidative stress and autophagy and their potential clinical applications. This is a timely and comprehensive review. There are nevertheless several issues that need to be addressed.
Response: The authors would like to thank the reviewer for the general comments regarding their manuscript. They considered all the issues that should be addressed, and they have enriched the revised manuscript.
Point 1: Subsection 4.2. It is necessary to elaborate on details of the interaction between autophagy and senescence.
Response 1: The authors appreciated this comment and enriched Subsection 4.2, elaborating on details of the interaction between autophagy and senescence.
Point 2: Reference 46 is not related to autophagy and senescence. It should be replaced with relevant references.
Response 2: The authors would like to thank the reviewer for this remark. They have replaced this Reference (Carroll et al. 2018) with a new one (Kwon et al. 2017) (Ref 36 in the revised manuscript).
Point 3: It is important to mention that the DNA damage repair system, autophagy, and senescence not only aid the process of purging ROS from the cellular environment but also deal with the consequences of ROS-mediated damage (The Introduction section, lines 60-63).
Response 3: The authors would like to thank the reviewer for this suggestion. They have added a related comment (“When the cell is required to confront these stressors, it has to its disposal an assortment of different mechanisms that not only attempt to encounter the consequences provoked by ROS in cellular molecules referred above, but also aid in the process of purging ROS from the cellular environment”) (lines 59-63 in the revised manuscript).
Point 4: Figure legends need to be expanded in order to enable the reader to understand figures without having to refer to the main text.
Response 4: The authors would like to thank the reviewer for the constructive suggestion. They have expanded the figure legends. The figure numbering has been changed. A new figure has been also added (Figure 5).
Point 5: The authors have not numbered the different sections and subsections properly (throughout the paper).
Response 5: The authors have taken into account this attentive remark. They have renumbered properly the different sections of the manuscript. They have re-arranged the sequence of some of the sub-sections and they have also added new parts in the revised manuscript.

Reviewer 2 Report
The authors present a very interesting review and detailed account on an important topic. They provide, a very well written, overview of cell senescence and autophagy in health and disease covering a wide range of background on both cell states. They also summarise the interrelation between senescence and autophagy providing the pathways involved in each and the importance of autophagy in cell homeostasis, but also how it acts as a perpetrator of cellular senescence.
However, it feels that this is only a report and when the reader is looking forward to read the suggestions on the therapeutic approaches/advances these are lacking. It feels kind of rushed on this part, which would make this review a step above any other out-there.
Specific comments:
1. Section 4 is overall well laid out, but it would benefit from more specific examples from the references that already used.
2. More summary figures or tables on section 4 on the therapeutic approaches are suggested if possible. Figures on the role of nanotechnology in autophagy, oxidative stress and cellular senescence would also be helpful if possible.
3. All figures would benefit from additional and more detailed (so the figure become a stand alone) captions. Not extensive ones but with a bit more detail.
4. SenTraGor is a very important chemical substrate for identifying lipofuscin in various type of cells from mesenchymal stem cells to cancer cells. In page 4, line 146, needs to better introduced, use "SenTraGorTM, firstly developed in the Gorgoulis lab in (year), is a biotinylated Sudan Black-B (SBB) chemical analog which reacts with high specificity and sensitivity against lipofuscin and is widely used for precise detection of senescent cells". Also need to include references about sentragor.
5. A read though (polishing) of the text is also needed. A few of the changes required in the text:
p1, line 32: Do not divide words with hyphen like senescence unless it is absolutely necessary. Same for tumorigenesis in page 3.
p1, line 32: use a comma (,) "in multiple tissues, but is also"
p1, line 39: The sentence "In this review there is an attempt to reveal the mechanistic aspects of this dual role." change to "This review is an attempt to reveal the mechanistic aspects of this dual role."
p1, line 40: Delete Moreover from the sentence.
p1, line 41-42: The sentence needs rewriting as it is confusing, does the "functioning as potential therapeutic targets" referees to ROS? If it does it needs to be therapeutic target. Please correct so it reads better.
p2, line 50: "When said proteins have fulfilled their purpose," change to these proteins.
p2, line 51-52: "...they are broken down to the building block they originated from and that is how the balance is kept in that context." it would read better if it would change to "they are broken down to the building blocks they originated from maintaining the balance in that context"
p2, line 89: "oxidative stress, acting act either preventing from ROS generation or controllably inducing" delete act and on "stress, acting on either preventing from ROS generation or controllably inducing"
p3, line108: "Senescent cells are defined by a general non-irreversible cell-cycle arrest" non-irreversible needs to be either irreversible or non-reversible.
p4, line 147-149: Sentence "Various theranostic applications might be developed based on identifying and targeting senescent cells [19] that are related to various pathologies" needs rewriting. Does this mean that theranostic applications will be targeting what exactly? Needs to give examples and be specific. Also need extra references with examples of senolytics that targeting senescent cells just referencing 19 is not enough you need more examples.
p10, line 443-446: Sentence needs references.
Author Response
Response to Reviewer 2 Comments
The authors would like to thank the reviewer for the valuable comments and for giving them the opportunity to revise their manuscript.
All the modifications have been highlighted in yellow in the revised version of the manuscript. The reference and figure numbering has also been changed.
Comments and Suggestions for Authors: The authors present a very interesting review and detailed account on an important topic. They provide, a very well written, overview of cell senescence and autophagy in health and disease covering a wide range of background on both cell states. They also summarise the interrelation between senescence and autophagy providing the pathways involved in each and the importance of autophagy in cell homeostasis, but also how it acts as a perpetrator of cellular senescence.
However, it feels that this is only a report and when the reader is looking forward to read the suggestions on the therapeutic approaches/advances these are lacking. It feels kind of rushed on this part, which would make this review a step above any other out-there.
Response: The authors would like to thank the reviewer for the general comments regarding their manuscript. They considered all the issues that should be addressed, and they have enriched the revised manuscript, specifically focusing on the therapeutic approaches/advances. Particularly, they have added new related sections (5. Therapeutic Approaches). They also highlighted the role of nanomaterials on inducing oxidative stress, as a therapeutic strategy. Thus, they slightly modified the previous title of subsection «6.1. Nanomaterials and Induction of Oxidative Stress» into «6.1. Nanomaterials and Induction of Oxidative Stress as a Therapeutic Strategy» and they have enriched the section 6 in the revised manuscript.
Point 1: Section 4 is overall well laid out, but it would benefit from more specific examples from the references that already used.
Response 1: The authors appreciated this comment. They have enriched section 4 and added a new part (5. Therapeutic Approaches). They have enriched section 6 with more specific examples. Particularly they have added examples based on the use of composite thermo-responsive titania-based materials, zinc oxide, silver, copper, and gallium nanoparticles as therapeutic agents. They have also added some more examples in the subsection 6.2 and also a new figure.
Point 2: More summary figures or tables on section 4 on the therapeutic approaches are suggested if possible. Figures on the role of nanotechnology in autophagy, oxidative stress and cellular senescence would also be helpful if possible.
Response 2: The authors would like to thank the reviewer for this remark. They have enriched the manuscript with more examples, and they have added a new figure in sub-section 6.2. Nanomaterials and Prevention from Oxidative Stress.
Point 3: All figures would benefit from additional and more detailed (so the figure become a stand alone) captions. Not extensive ones but with a bit more detail.
Response 3: The authors would like to thank the reviewer for this suggestion. They have expanded the figure legends in order to stand alone. The figure numbering has been changed in the revised manuscript.
Point 4: SenTraGor is a very important chemical substrate for identifying lipofuscin in various type of cells from mesenchymal stem cells to cancer cells. In page 4, line 146, needs to better introduced, use "SenTraGorTM, firstly developed in the Gorgoulis lab in (year), is a biotinylated Sudan Black-B (SBB) chemical analog which reacts with high specificity and sensitivity against lipofuscin and is widely used for precise detection of senescent cells". Also need to include references about sentragor.
Response 4: The authors would like to thank the reviewer for the constructive suggestion. They have introduced SenTraGor, mentioning related studied that have exploited its potential in sub-section 2.3. Liposfuscin detection.
Point 5: A read though (polishing) of the text is also needed. A few of the changes required in the text: Response 5: The authors have taken into account this attentive remark. They have taken into account almost all of the reviewer’s comments. p1, line 32: Do not divide words with hyphen like senescence unless it is absolutely necessary. Same for tumorigenesis in page 3. Response: The authors understand the reviewer’s comment, but they cannot control this issue, since the words are divided with hyphen due to the proposed template of the journal that they have used. Some of the words that the reviewer has mentioned (e.g., line 32) have automatically corrected, due to the addition of some words in the same line. But this is not controllable throughout the whole manuscript. p1, line 32: use a comma (,) "in multiple tissues, but is also" Response: The authors have used a comma, as the reviewer suggested. p1, line 39: The sentence "In this review there is an attempt to reveal the mechanistic aspects of this dual role." change to "This review is an attempt to reveal the mechanistic aspects of this dual role." Response: The authors have replaced the sentence "In this review there is an attempt to reveal the mechanistic aspects of this dual role." with "This review is an attempt to reveal the mechanistic aspects of this dual role.", as the reviewer suggested. p1, line 40: Delete Moreover from the sentence. Response: The authors have deleted “Moreover” form this sentence. p1, line 41-42: The sentence needs rewriting as it is confusing, does the "functioning as potential therapeutic targets" referees to ROS? If it does it needs to be therapeutic target. Please correct so it reads better. Response: The authors understood that comment and replaced the word “targets” with that of “agents” to clarify the potential use of them in therapeutics. p2, line 50: "When said proteins have fulfilled their purpose," change to these proteins.Response: The authors have changed the phrase "When said proteins… " to “These Proteins…”. p2, line 51-52: "...they are broken down to the building block they originated from and that is how the balance is kept in that context." it would read better if it would change to "they are broken down to the building blocks they originated from maintaining the balance in that context" Response: The authors have changed the phrase "...they are broken down to the building block they originated from and that is how the balance is kept in that context." to "they are broken down to the building blocks they originated from maintaining the balance in that context". p2, line 89: "oxidative stress, acting act either preventing from ROS generation or controllably inducing" delete act and on "stress, acting on either preventing from ROS generation or controllably inducing" Response: The authors have changed the phrase "oxidative stress, acting act either preventing from ROS generation or controllably inducing" to " oxidative stress, acting on either preventing from ROS generation or controllably inducing". p3, line108: "Senescent cells are defined by a general non-irreversible cell-cycle arrest" non-irreversible needs to be either irreversible or non-reversible. Response: The authors have changed non-irreversible to non-reversible. p4, line 147-149: Sentence "Various theranostic applications might be developed based on identifying and targeting senescent cells [19] that are related to various pathologies" needs rewriting. Does this mean that theranostic applications will be targeting what exactly? Needs to give examples and be specific. Also need extra references with examples of senolytics that targeting senescent cells just referencing 19 is not enough you need more examples. Response: The authors understand that perhaps the meaning of this sentence was not clear enough and modified it, by summarizing some of the previously mentioned information about the senescence markers. They have also added extra references. p10, line 443-446: Sentence needs references. Response: The authors have added a new reference at these lines (lines 524-528 in the revised manuscript).

Reviewer 3 Report
Manuscript of P. Pantelis et al. “The dual role of oxidative stress-induced autophagy in cellular senescence: Comprehension and Therapeutic Approaches” is devoted to the problem of interrelashionship between senescence, oxidative stress and autophagy.
The authors consider the role of oxidative stress and its mediating mechanisms in development of senescence and aging. The possibilities of autophagy regulation under oxidative stress and the mechanisms mediating the relationship between autophagy and senescence are also considered. A separate section is devoted to the role of nanotechnology in the regulation of autophagy, oxidative stress and senescence. The review contains a number of interesting data on the issues under consideration, but in its present form, it is not ready for publication and needs a thorough and radical revision.
Major comment.
1. It is very desirable to improve the review structure.
Possible recommendation looks as follows:
Following Introduction, the sections could follow providing information about the concepts that are the subject of the review:
-Cellular senescence
-Oxidative Stress
-Autophagy.
Then authors give information about relationship between these concepts:
-Autophagy and cellular senescence
-Pathways activating autophagy by ROS
-The role of Nanotechnology in Autophagy, Oxidative Stress and Cellular Senescence.
This sequence is advisory in nature and can be changed by the authors; its main purpose is to facilitate the reader's perception of the review. Otherwise, the authors' presentation of the text looks inconsistent.
2. Figure 2 fig. 2 needs to be corrected.
- If ROS can inhibit ATG4? Yes, because Atg4 is a cysteine proteinase.
- If ATG4 inhibits LC3 II? It is poorly correct to ascribe LC3 inhibition to Atg4. Indeed, this factor proteolytically processes LC3/Atg8, initiating its ultimate conversion into active form conjugated with phosphatidylethanolamine. Further, Atg4 exerts delipidation of LC3, with resumption of cyclic LC3 involvement into autophagy.
In this relation, see line 219: why “ROS-mediated Atg4 inactivation provokes LC3-II accumulation”?
- LC3-II inhibits mTORC1? Most probably, not (however, author could have some information supporting another point of view), but rather the other way round. In general, brunch “ROS-Atg4-LC3II” should be placed in the scheme represented regardless of mTORC1.
In general, this figure (and its description) should be placed after “Autophagy” section, and then everything becomes clear.
3. A lot of work is needed on the references to the text.
Discrepancies found by the reviewer:
Line 267, 269 – reference {34} does not correspond to text description
Line 375 - reference {50} does not correspond to text description.
Line 377 - reference {19} does not correspond to text description
Line 391 - reference {52,53} do not correspond to text description
Line 126 [11,11]. SASP mediates – double citing of ref. Nr11
The authors need to edit the list of citations and bring it in line with the text
Minor comments.
Misprints in the text noticed by reviewer
1 Line 174 Oxygen is often released after the -oxidation of lipids
2 Line 190 -tocopherol
3 Line 230 By blocking the mTORC1 pathway,…..
Line 235 The mammalian homologous protein mTOR may form two complexes, the 1 and 2 mTOR complexes, with differing functional characteristics. THIS EXPLAINING SENTENCE SHOULD PRECEDE THAT OF LINE 230.
4 Line 236, 251 protein raptor
Line 237 Rictor
Authors should choose capital/uppercase writing of the factors mentioned in the review.
5 Line 284 of the ,
6 Line 303 Another significant “method”=rather “pathway”

Author Response
Response to Reviewer 3 Comments
The authors would like to thank the reviewer for the valuable comments and for giving them the opportunity to revise their manuscript.
All the modifications have been highlighted in yellow in the revised version of the manuscript. The reference and figure numbering has also been changed.
Comments and Suggestions for Authors: Manuscript of P. Pantelis et al. “The dual role of oxidative stress-induced autophagy in cellular senescence: Comprehension and Therapeutic Approaches” is devoted to the problem of interrelashionship between senescence, oxidative stress and autophagy.
The authors consider the role of oxidative stress and its mediating mechanisms in development of senescence and aging. The possibilities of autophagy regulation under oxidative stress and the mechanisms mediating the relationship between autophagy and senescence are also considered. A separate section is devoted to the role of nanotechnology in the regulation of autophagy, oxidative stress and senescence. The review contains a number of interesting data on the issues under consideration, but in its present form, it is not ready for publication and needs a thorough and radical revision.
Response: The authors would like to thank the reviewer for the general comments regarding their manuscript. They considered all the issues that should be addressed, and they have enriched the revised manuscript.
Major comment. 1. It is very desirable to improve the review structure. Possible recommendation looks as follows: Following Introduction, the sections could follow providing information about the concepts that are the subject of the review: -Cellular senescence -Oxidative Stress -Autophagy. Then authors give information about relationship between these concepts: -Autophagy and cellular senescence -Pathways activating autophagy by ROS -The role of Nanotechnology in Autophagy, Oxidative Stress and Cellular Senescence. This sequence is advisory in nature and can be changed by the authors; its main purpose is to facilitate the reader's perception of the review. Otherwise, the authors' presentation of the text looks inconsistent. Response: The authors would like to thank the reviewer for this suggestion. They have re-arranged the sub-sections of the manuscript. 2. Figure 2 fig. 2 needs to be corrected. Response: The authors would like to thank the reviewer for this remark. They have corrected Figure 2.
- If ROS can inhibit ATG4? Yes, because Atg4 is a cysteine proteinase. - If ATG4 inhibits LC3 II? It is poorly correct to ascribe LC3 inhibition to Atg4. Indeed, this factor proteolytically processes LC3/Atg8, initiating its ultimate conversion into active form conjugated with phosphatidylethanolamine. Further, Atg4 exerts delipidation of LC3, with resumption of cyclic LC3 involvement into autophagy. In this relation, see line 219: why “ROS-mediated Atg4 inactivation provokes LC3-II accumulation”? Response: The authors would like to thank the reviewer for this suggestion. They have modified this part of the manuscript, enhancing the semantic flow of understanding (lines 319-323 in the revised manuscript). - LC3-II inhibits mTORC1? Most probably, not (however, author could have some information supporting another point of view), but rather the other way round. In general, brunch “ROS-Atg4-LC3II” should be placed in the scheme represented regardless of mTORC1. Response: The authors would like to thank the reviewer for this remark. They have corrected Figure 2. In general, this figure (and its description) should be placed after “Autophagy” section, and then everything becomes clear. Response: The authors are thankful for this comment. They have placed this figure after «Autophagy» sub-section. 3. A lot of work is needed on the references to the text. Discrepancies found by the reviewer: Line 267, 269 – reference {34} does not correspond to text description. Response: The authors are thankful for this comment and have replaced this reference with a new one (Xinbing et al. 2014) (Ref No 55 in the revised manuscript). Line 375 - reference {50} does not correspond to text description. Response: The authors are thankful for this comment. They have corrected this reference (Ref No 34 in the revised manuscript). Line 377 - reference {19} does not correspond to text description. Response: The authors are thankful for this comment and have replaced this reference with a new one (Park et al. 2013) (Ref No 40 in the revised manuscript). Line 391 - reference {52,53} do not correspond to text description. Response: The authors are thankful for this comment. They have corrected these references (Ref No 42-44 in the revised manuscript). Line 126 [11,11]. SASP mediates – double citing of ref. Nr11. Response: The authors are thankful for this remark and have deleted the double reference. (Ref No 12 in the revised manuscript). The authors need to edit the list of citations and bring it in line with the text. Response: The authors are thankful for this suggestion. They have edited the reference list, they have added new ones and they have brought them in line with the text.
Minor comments. Misprints in the text noticed by reviewer. 1 Line 174 Oxygen is often released after the -oxidation of lipids. Response: The authors would like to thank the reviewer for this remark. They have corrected this misprint (line 179 in the revised manuscript). 2 Line 190 -tocopherol. Response: The authors would like to thank the reviewer for this remark. They have corrected this misprint (line 194 in the revised manuscript). 3 Line 230 By blocking the mTORC1 pathway,….. Line 235 The mammalian homologous protein mTOR may form two complexes, the 1 and 2 mTOR complexes, with differing functional characteristics. THIS EXPLAINING SENTENCE SHOULD PRECEDE THAT OF LINE 230. Response: The authors would like to thank the reviewer for this suggestion. They have updated this part of the manuscript (lines 333-336 in the revised manuscript). 4 Line 236, 251 protein raptor Response: The authors would like to thank the reviewer for this suggestion. They have corrected this misprint (lines 338, 340 in the revised manuscript). Line 237 Rictor Response: The authors would like to thank the reviewer for this suggestion. They have corrected this misprint (lines 341 in the revised manuscript). Authors should choose capital/uppercase writing of the factors mentioned in the review. Response: The authors would like to thank the reviewer for this suggestion. They have considered it. 5 Line 284 of the , Response: The authors would like to thank the reviewer for this suggestion. They have corrected this misprint (line 393 in the revised manuscript). 6 Line 303 Another significant “method”=rather “pathway” Response: The authors would like to thank the reviewer for this suggestion. They have corrected this misprint (line 413 in the revised manuscript).
